# Compartmentalised mucosal and blood immunity to SARS-CoV-2 is associated with high seroprevalence before the Delta wave in Africa

## Abstract

**Background** The reported number of SARS-CoV-2 cases and deaths are lower in Africa compared to many high-income countries. However, in African cohorts, detailed characterisation of SARS-CoV-2 mucosal and T cell immunity are limited. We assessed the SARS-CoV-2-specific immune landscape in The Gambia during the presence of the pre-Delta variant in July 2021.

**Methods** A cross-sectional assessment of SARS-CoV-2 immunity in 349 unvaccinated individuals from 52 Gambian households was performed between March–June 2021. SARS-CoV-2 spike (S) and nucleocapsid (N) specific binding antibodies were measured by ELISA, variant-specific serum neutralizing-antibodies (NAb) by viral pseudotype assays and nasal fluid IgA by mesoscale discovery assay. SARS-CoV-2 T-cell responses were evaluated using ELISpot assay.

**Results** We show that adjusted anti-Spike antibody seroprevalence is 56.7% (95% confidence interval (CI) 49.0-64.0), with lower rates in children <5 years (26.2%, 13.9-43.8) and 5-17 years (46.4%, 36.2-56.7) compared to adults 18-49 years (78.4%, 68.8–85.8). Among spike-seropositive individuals, NAb titres are highest against Alpha variant (median IC50 110), with 27% showing pre-existing Delta variant titres >1:50. T-cell responses are higher in spike-seropositive individuals, although 34% of spike-seronegative individuals show responses to at least one antigen pool. We observe strong correlations within SARS-CoV-2 T-cell, mucosal IgA, and serum NAb responses.

**Conclusions** High SARS-CoV-2 seroprevalence in The-Gambia induce mucosal and blood immunity, reducing Delta and Omicron impact. Children are relatively protected from infection. T-cell responses in seronegative individuals may indicate either pre-pandemic cross-reactivity or individuals with a T-cell dominated response to SARS-CoV-2 infection with absent or poor humoral responses.

## Plain language summary

The COVID-19 pandemic caused many illnesses and deaths worldwide. In Africa, the reported number of COVID-19 cases remained low and while under-reporting may have played a role, no increase in deaths was seen in The Gambia during the first year of the COVID-19 pandemic. There is limited data on how the immune system of Africans respond to COVID-19 infections. Our aim was to assess COVID-19 immune responses using blood and nasal swabs collected from the participants. Measurement of immune cell function and antibody levels, showed that COVID-19 infection activated different specialized cells of the immune system, including certain white blood cells (T cells) and antibodies. Antibodies are proteins made by the cells in our body, found in the blood and mucous membranes and protect against infection. We also found that COVID-19 infection rates were higher in adults compared to children and that the immune responses from an infection with earlier COVID-19 variants may have conferred protection against infection with later variants. These findings contribute to our understanding of how Africans of different ages responded to COVID-19 infection.

SARS-CoV-2 infection was declared a pandemic by the World Health Organisation (WHO) on 11th March 2020 and as of 07th July 2024, an estimated 775,673,955 confirmed cases and a total of 7,053,524 related deaths have been recorded worldwide[1]. Africa, which accounts for 16% of the world population, has recorded the lowest rates of SARS-CoV-2 infections with only 9,580,532 million reported cases and 175,510 related deaths so far in a total of 54 countries, accounting for only 1.2% of total cases worldwide[1].

✉e-mail: ya-jankey.jagne@lshtm.ac.uk; beate.kampmann@charite.de

In The Gambia, the first case of COVID-19 was recorded on the 17th of March 2020 and as of 07th of July 2024, 12,627 individuals have been confirmed infected with the virus, with 372 related deaths recorded[1,2]. As seen with other countries, a series of nationwide lockdowns were implemented to curb the spread of the virus. A state of emergency, which included lockdown of the border between Gambia and Senegal, all schools, some workplaces, places of worship and nonessential shops, was announced on 27th March 2020, and this was further extended multiple times[3,4].

The low number of reported SARS-CoV-2 cases and deaths in The Gambia is consistent with existing data from most African countries showing lower rates of infection and mortality in Africa compared to many high-income countries. The reported COVID-19-related deaths remained low in most African countries even during SARS-CoV-2 Alpha and Delta variant waves, despite the higher disease severity observed in many high-income countries. Although under-reporting and the young population structure may have played a role in the number of recorded cases seen, the resource-limited care systems in many African countries were not overwhelmed, suggesting that many cases may have been mild or asymptomatic[4]. No excess mortality was observed in The Gambia during the first year of the COVID-19 pandemic[5]. This milder disease may also be due to a higher prevalence of cross-reactive SARS-CoV-2 antibodies and T-cell responses from previous exposure to other coronaviruses[6].

The objective of our study was to describe the seroprevalence and immunity to SARS-CoV-2 in The Gambia after the first two waves, prior to the introduction of any SARS-CoV-2 vaccines and the emergence of the Delta and Omicron variants. We hypothesised that substantial humoral and cellular immunity would have been present in the population due to high attack rates with earlier SARS-CoV-2 variants, potentially limiting the impact of Delta and Omicron variants that followed in the third and fourth SARS-CoV-2 waves in The Gambia. While many studies from Africa have measured binding and neutralising antibody responses, very few have established the extent of mucosal antibodies and T-cell immunity, which may both play a significant role in protection from future infections and severe disease. We use a combination of immunological assays to measure antibody levels and immune cell function in our participants. High SARS-CoV-2 exposure in The Gambia results in strong immune responses in the nose and blood, which help in limiting the effects of the Delta and Omicron variants. Children tend to be less affected by the virus. Interestingly, some individuals without detectable antibodies still show antigen-specific T-cell responses, possibly due to previous infections. These findings suggest that T-cell assays are important to include in epidemiological studies, in addition to antibody assays.

## Methods

### Participant recruitment and data collection

Following community and household sensitisation, interested families were invited to the MRCG clinical site for consenting and screening. Community sensitisation involved meeting influential individuals such as religious leaders within the community to sensitise them on the research project and ensure that they understand and are supportive of the trial. Individual households were then visited to inform family members about the project. Families that agree to be part of the study are invited to the MRCG clinical site for consent and screening. In total, 349 participants from 52 households in the West Coast Region and the Kanifing Municipal Area of The Gambia were recruited.

Household participants with travel planned during the study follow-up period were excluded. Written consent or a thumbprint was obtained from all participants above 18, while assent was obtained from participants aged 12–18. Consent was obtained from the parents or guardians for participants under 12. Study participants were not financially compensated but provided with free transportation during clinical visits.

Study approval was given by the joint Gambia Government and Medical Research Council Unit, The Gambia (MRCG) Ethics committee and the London School of Hygiene and Tropical Medicine ethics committee (LEO project ID 22556). The protocol was registered at clinicaltrials.gov

(Transmission of Respiratory Viruses in Household in The Gambia: TransVIR; clinicaltrials.gov NCT05952336)[7].

### Sample collection and processing

During the baseline visit, venous blood was collected for peripheral blood mononuclear cell isolation and cryopreservation, and serum separation and storage. A synthetic absorptive matrix (SAM) strip was used to collect nasal lining fluid sample. In addition, a combined throat and nose swab using flocked swabs was collected and transferred into RNAprotect cell reagent (Qiagen)[8].

### Pre-pandemic samples

Following ethical approval from the Gambia Government/MRCG Joint Ethics Committee, 825 biobanked sera collected from unrelated studies conducted in 2016 in three villages of West Kiang Lower River Region by the MRC Unit, The Gambia (SCC 1269, SCC 1449, SCC 1185), were used to determine pre-pandemic cross-reactivity with SARS-CoV-2 antigens and establish the specificity of the antibody-ELISA. The need for reconsenting participants was waived by the Ethics committee as all original participants had agreed to the storage of their remaining samples for future research. The samples allowed for a representative spread across age groups (1–75 years), seasons and sex. Twenty samples were randomly selected from males and females for each month for two years to account for potential seasonal variations to exposure to human coronaviruses.

### SARS-CoV-2 ELISA

Immunlon 4 HBX 96-well microplates (Thermo Scientific) were coated overnight at 4 °C with either SARS-CoV-2 full-length untagged nucleo-capsid (Uniprot ID P0DTC9 (NCAP_SARS2)) produced in E. coli at 2 µg/mL or full-length spike protein extracellular domain (amino acid 14-1213) with replacements on the furin cleavage site R684-R689 and K986-V987 by a single alanine residue and PP from mammalian cells at 0.9 µg/mL[9,10].

Plates were washed three times (PBS-T) and blocked for 1 hour with assay buffer (1:1 Casein blocking buffer: Dulbecco Phosphate Buffer Saline (DPBS). Heat-inactivated samples (56 °C, 30 min) were diluted 1:200, with WHO-standard calibrators (NIBSC 20/136) used to generate a 12-point curve (1.96–976.32 BAU/mL). Each plate included blank, positive and negative controls. Each plate included blank, positive and negative controls.

After a 2-hour incubation at 25 °C, plates were washed and incubated with HRP-conjugated goat anti-human IgG (1:500, Invitrogen) for 1 hour. Detection used SureBlue TMB (10 min), and absorbance was read at 450 nm (Multiskan-Go spectrophotometer-ThermoFisher). Binding antibody units (BAU) were calculated using the elisa-dolittle Python programme (*GitHub: lindseyb101112/elisa-dl*). Sample OD values were interpolated against plate-specific curves. Serostatus was determined using a three-ratio cut-off method, using mean OD values from 50 standard curves.

### SARS-CoV-2 lentiviral pseudotype neutralisation assays

**Cell culture.** HEK293T cells were cultured in DMEM, and CHO cells in Ham's F-12 medium, both supplemented with 10% fetal bovine serum and 1% penicillin/streptomycin. Cells were maintained at 37 °C with 5% $CO_2$ and passaged thrice weekly using PBS washes and trypsin-EDTA for detachment. HEK293T and CHO cells were routinely checked for mycoplasma contamination. To maintain selection, CHO cells were cultured with 2 µg/mL Puromycin and 100 µg/mL Hygromycin B every other passage. Toxilight bioassays (Lonza) were periodically performed on the cell lines, which were also transduced with VSV-G enveloped vectors. Cell lines were not authenticated.

**SARS-CoV-2 lentiviral pseudotype generation and titration.** Pseudotyped viruses (PVs) were generated as described by Di Genova et al.[11]. HEK293T cells were co-transfected with lentiviral Gag-Pol (p8.91), a luciferase reporter (pCSFLW) and SARS-CoV-2 Spike (Wuhan, B.1.1.7, B.1.617.2, BA.1, BA.2) in a pcDNA3.1 vector using FuGENE HD. After

18 h, fresh DMEM was added, and 30 h later, PV-containing supernatants were filtered (0.45 μM), aliquoted, and stored at −80 °C.

For titration, PVs were serially diluted and incubated with CHO cells expressing ACE2/TMPRSS2 in 96-well plates for 48 h. Cells were lysed with Bright-Glo luciferase reagent and luminescence (RLU) was measured using a GloMax luminometer.

**Pseudotype neutralisation assays (PtNA).** Neutralisation assays followed Di Genova et al.[11]. Human serum was serially diluted (1:20 to 1:4860) in Ham's F-12 and incubated with PVs (~$5 \times 10^6$ RLU/well) for 1 h at 37 °C. CHO-ACE2/TMPRSS2 cells (20,000/well) were then added and incubated for 48 h before luciferase assay. Internal calibrants and control wells (cell-only and serum-free) were included. $NT_{50}$ values below 1:50 were considered negative.

**SAM strips processing for mucosal IgA measurements.** SAM strips were thawed in their containers on ice for 30 minutes, detached using disposable forceps and placed into a tube containing 300 μl of elution buffer containing 1% Bovine Serum Albumin (BSA) (MELFORD, 9048-46-8), 1× protease inhibitor cocktail (Calbiochem, 539131) and PBS (Gibco, 10010031). The tube containing SAM strip and buffer is vortexed for 30 seconds, then transferred into spin columns and centrifuged at $16,000 \times g$, for 20 minutes at 4 °C. The eluate contained within the spin tube was transferred into labelled cryotubes and heat-inactivated at 56 °C for 30 min. Samples were then used to measure mucosal IgA antibodies.

**Meso-scale discovery assay to measure mucosal IgA responses in nasal lining fluid samples.** Mucosal anti-spike IgA responses to different SARS-CoV-2 variants (Ancestral, B.1.1.7, B.1.617.2, BA.1, BA.2) were measured using a V-PLEX SARS-CoV-2 Panel 25 (IgA) Kit (K15585U, MSD, Maryland, USA) according to the manufacturer's instructions. To measure IgA binding antibodies, 96-well plates were blocked with Blocker A solution for 30 minutes. Plates were washed with MSD Wash buffer, and 50 μl of nasal mucosal lining fluid samples obtained from the SAM strips were diluted 1:25–1:800 and added to wells, along with standards and controls and incubated for 2 hours. After a 2-hour incubation and a wash step, diluted detection antibody (MSD SULFO-TAG™ Anti-Human IgA Antibody, was added and plate incubated for 1-hour. Plate was washed after the incubation and MSD GOLD Read Buffer B added. Plates were read using the Discovery Bench 4.0 software provided by MESO SCALE DIAGNOSTICS, LLC.

**SARS-CoV-2 T-cell IFN-gamma ELISpots.** S, N and M-specific T-cell responses were quantified using a Human IFN-gamma ELISPOT kit (Mabtech) following the manufacturer's instructions and published protocols[12]. Briefly, 50 μl of $2.5 \times 10^5$ PBMCs/well were stimulated overnight (~18 h) with peptides representing S1, S2, M and N proteins (15-mers overlapping by 10 amino acids) at a final concentration of 2 μg/ml, a peptide pool of CMV, EBV and Influenza peptides Stem Cell Technologies, USA). Staphylococcal Enterotoxin B/Phytohemagglutinin-L (SEB/PHA) at 2 μg/ml (Sigma-Aldrich, USA) and medium alone were used as positive and negative controls, all in triplicates. The numbers of SARS-CoV-2-specific IFN-gamma-secreting T cells were determined using an AID Elispot Reader (AID, Strasberg, Germany) and analysed using ELISpot 7.0 Software (AID).

Results were expressed as IFN-gamma spot forming units (SFU)/$10^6$ PBMC, after subtracting the responses from the medium control well. The results for a sample were considered for further analysis when responses in the positive control well were higher than 200 IFN-gamma SFU/$10^6$ PBMCs and negative control lower than 50 IFN-gamma SFU/$10^6$ PBMC cells. Antigen wells with responses higher than mean plus 2 standard deviations of negative control wells were considered to have a positive T-cell response.

**Statistical analysis.** All data management, visualisation and statistical analyses were performed in R (v4.3.1) and RStudio (2023.09.1) using

various packages, including utils (v1.4.3), dplyr (v1.1.3), ggplot2 (v3.4.4), reshape2 (v1.4.4), ggpubr (v0.6.0), ggbreak (v.0.1.2), FSA (v0.9.5), corrplot (v.0.92), dunn.test (v1.3.5), forcats (v1.0.0), wesanderson (v0.3.6), tidyverse (v2.0.0), scales (v1.2.1), stringr (v1.5.0), ggsignif (v0.6.4), cowplot (v1.1.1), viridis (v0.6.5), Hmisc (v5.1–2), patchwork (v1.1.3) and stats (v.4.3.1).

Seroprevalence estimates accounted for household clustering using a mixed-effects logistic regression model with household as a random effect. The Wilcoxon test compared responses between serogroups and SARS-CoV-2 antigens for antibody and T-cell data. Spearman correlations assessed relationships between T-cell, neutralising and mucosal antibody responses, with adjusted $P$ values calculated using the Holm-Sidak method.

Cellular and T-cell responses across age groups were analysed using the Kruskal–Wallis test, followed by Dunn's post hoc test for significant differences. Multiple comparisons were corrected using the Bonferroni–Holm method. Statistical significance was set at $P < 0.05$.

## Results

### Cohort and participants

Between 2nd March 2021 and 13th June 2022, 349 participants from 52 households were recruited into the study, prior to the Delta wave[3]. The data reported here represent immunological data generated from the baseline sample collected in the study, which was a prospective, longitudinal, observational household cohort study of SARS-CoV-2 incidence and immunity (clinicaltrials.gov NCT05952336). The longitudinal SARS-CoV-2 incidence data from the study are reported elsewhere[8]. The Gambia is a small country in West Africa with a sub-tropical climate, ranked 174th by the United Nations Human Development Index in 2021[13]. The study was conducted at two urban sites, the West Coast Region and Kanifing Municipality of The Gambia. A median of six individuals were recruited from each household (IQR 5–8), with 41 participants under 5 years old, 153 between 5–17 years old, 130 adults aged 18–49 years old, and 25 adults aged ≥50 years old. Two hundred and one (57.6%) participants were female. No participants had received SARS-CoV-2 vaccines at the time of study recruitment.

### Pre-pandemic serological cross-reactivity and SARS-CoV-2 seroprevalence in 2021

Biobanked serum samples ($n = 825$) collected in 2016 from three villages in West Kiang, The Gambia, were used to assess pre-pandemic cross-reactivity to SARS-CoV-2 antigens. These included samples from 69 children <5 years, 240 aged 5–17, 239 aged 18–49, and 277 aged ≥50 years. Analysis of binding antibody responses to SARS-CoV-2 spike (S) and nucleocapsid (N) in pre-pandemic serum showed that 76.4% (630/825) were negative for both antigens (S−N−, Fig. 1a). Cross-reactivity to N alone (S−N+) was observed in 19.2% (158/825), while only 3.4% (28/825) were reactive to spike (S+N−) and 1.1% (9/825) to both antigens (S+N+). Nucleocapsid cross-reactivity increased significantly with age: 4.3% (3/69) in <5 years, 9.6% (23/240) in 5–17 years, 23.8% (57/239) in 18–49 years and 30.3% (84/277) in ≥50 years (Fig. S1; <5 years and 18–49 years; $p = 0.0005$; 5–17 years and 18–49 years $p = 0.0002$; 5–17 years and 50+years, $p = 2.56^{e-08}$; <5 years and 50+ years, $p = 2.95^{e-06}$). In contrast, spike cross-reactivity showed no association with age.

Following the first two pandemic waves, 43.1% (150/348) of participants were seropositive to both antigens (S+N+), 13.2% (46/348) to S+N− and 39.4% (138/348) seronegative (S−N−) to both antigens (Fig. 1b). Spike antibody titres were significantly higher in S+N+ individuals than in S+N− ($p < 0.0001$, Fig. 1b). Similarly, N titres were higher in S+N+ than S−N+ individuals ($p = 0.0007$, Fig. 1b). No such titre differences were observed among pre-pandemic samples (Fig. 1a).

Due to the high pre-pandemic anti-N cross-reactivity observed, adjusted seroprevalence estimates were based on reactivity to spike alone (a combination of S+N+ and S+N− groups). The overall seroprevalence in the cohort prior to the Delta wave, adjusted for household clustering, was 56.7% (95% CI 49.0–64.0). This estimate varied by age; seroprevalence was

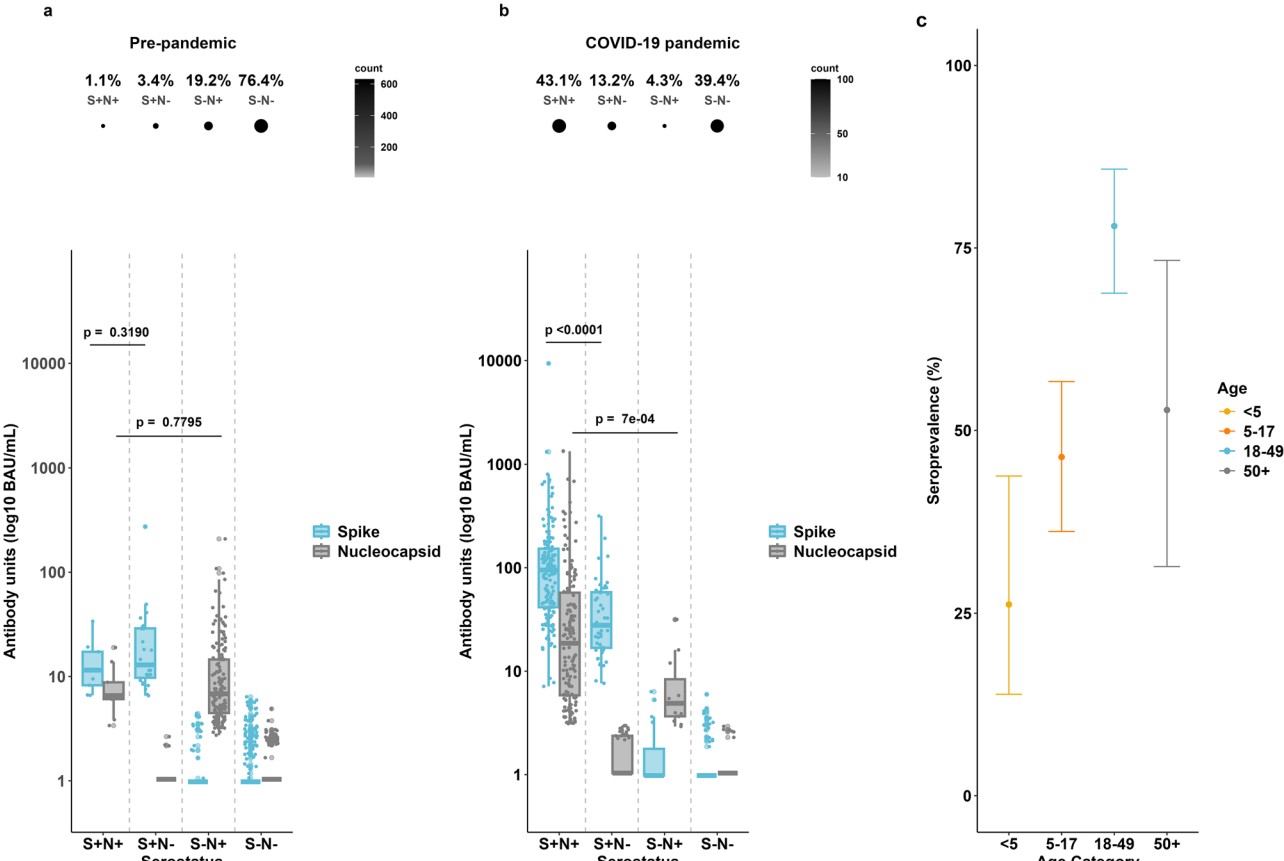

**Fig. 1 | Antibody responses to SARS-CoV-2 spike (S) and nucleocapsid (N).**
Participants are grouped by reactivity to both antigens (serostatus = S+N+, S+N−,
S−N+ and S−N−). **a** The proportion of participants within each serostatus group
(upper panel) and quantitative spike and binding nucleocapsid antibody (lower
panel) in pre-pandemic sera ($n = 825$). **b** The proportion of participants within each
serostatus group (upper panel) and quantitative spike and binding nucleocapsid
antibody (lower panel) following the first two SARS-CoV-2 waves ($n = 349$).

Boxplots depict means with 95% confidence intervals. BAU/mL = binding antibody
units/mL. Wilcoxon rank sum test was used to compare responses between S anti-
body responses in S+N+ and S+N− groups, as well as N antibody responses in S+N
+ and S−N+ groups. **c** Seroprevalence estimates for each age category based on
spike reactivity, adjusted for household clustering. The vertical error bars indicate
the 95% confidence intervals (CIs) of the estimates for the different age categories: <5
years, 5–17 years, 18–49 years and 50+ years.

only 26.2% (95% CI 13.9–43.8) in children aged under 5 years, 46.4% (95%
CI 36.2–56.7) in participants aged 5–17 years, 78%, (95% CI 68.8–85.8) in
those aged 18–49 and 52.8% (95% CI 31.4–73.3) in participants aged ≥50
years (Fig. 1c).

### Neutralising antibody responses to SARS-CoV-2 variants
Serum neutralising antibody (NAb) responses to SARS-CoV-2 variants that
circulated by the time of sampling (ancestral and Alpha) and those that
emerged after (Delta, Omicron BA.1 and BA.2) were measured using a PtNA
assay[14] (Fig. 2a). Participants seropositive for spike antibodies (S+N+ and S
+N−) had higher NAb titres to ancestral and Alpha variants than those that
were seronegative (S−N+ and S−N−), with the highest titres seen in S+N+
participants (Fig. 2b). Among spike-seropositive participants, NAb titres were
highest against the Alpha variant (median IC50 S+N+: 211.1, S+N−:96.5),
followed by the ancestral variant (median IC50 S+N+:117.0, S + N:32.9).
Age did not influence the NAb levels for most variants, except for the
ancestral strain, where 5–17-year-olds (median IC50 = 118.0, 95% CI =
3.8−974.0) had higher responses than those 18–49 years (median IC50 =
69.0, 95% CI = 10.0−480.0) (adjusted $p$ value = 0.014; Supplementary Fig. 2).

### Mucosal IgA responses to SARS-CoV-2 variants
SARS-CoV-2 variant-specific binding IgA was measured in nasal lining
fluid, focused on the same variants as above. Spike-specific mucosal IgA
titres were highest in participants who were seropositive to both spike and
nucleocapsid (S+N+) across all variants (Fig. 3a). Participants who were

only seropositive to spike (S+N−) had slightly lower titres than those S+N
+ that were still higher for ancestral, Alpha and Delta variants compared to
seronegative (S−N−) individuals (Fig. 3b). Mucosal IgA responses were
independent of age for all variants (Supplementary Fig. 3).

### T-cell responses to SARS-CoV-2 antigens
Interferon-gamma ELISpot assays were performed on participants with
adequate peripheral blood mononuclear cell (PBMC) recovery ($n = 289$) to
assess T-cell responses to SARS-CoV-2 S, N and M proteins. Following
exclusion due to high background ($n = 23$), data from 266 participants were
analysed. T-cell responses to all pools tested (S1, S2, N, M) were higher in
spike-seropositive (S+N+ or S+N−) individuals compared to those ser-
onegative for both spike and nucleocapsid (S−N−; Fig. 4). Interestingly,
33% of the S+N+ group and 43% of the S+N− had no reactivity to any
T-cell pools. However, and unlike the low serum NAb activity observed in
the S−N− group (Fig. 2), several spike and nucleocapsid seronegative
individuals had high T-cell responses, in some cases to multiple antigens
(Fig. 4a, c). T-cell responses in seropositive participants aged 5–17 years
were lower than those observed in participants aged 18–49 years for several
SARS-CoV-2 antigen pools (S1-adjusted $p$ value = 0.0103, $M$-adjusted $p$
value = 0.007, $N$-adjusted $p$ value = 0.0006; Supplementary Fig. 4).

### Correlation between different immune responses
To assess whether blood and mucosal immune responses were correlated
with each other or compartmentalised, a correlation matrix was constructed

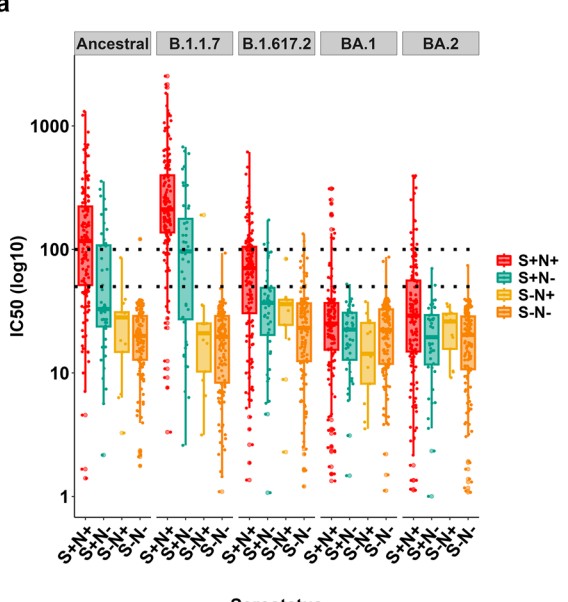

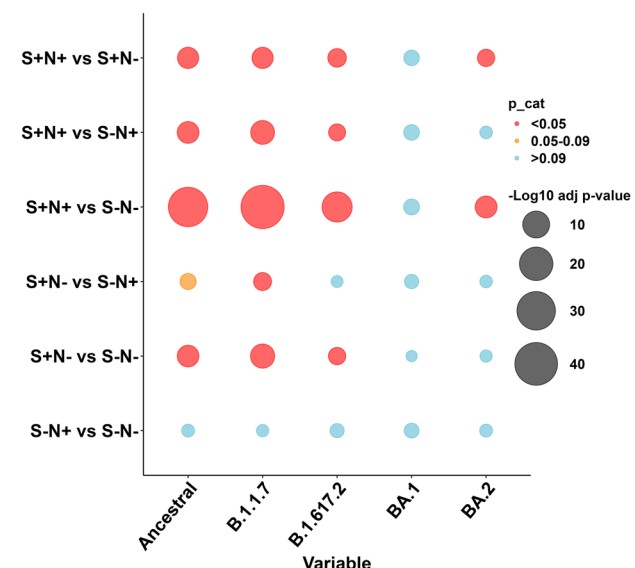

**Fig. 2 | Neutralising antibody (NAb) responses to SARS-CoV-2 variants.**
**a** Reciprocal IC50 (log10) serum titres to Ancestral, Alpha (B.1.1.7), Delta (B.1.617.2), Omicron BA.1 and BA.2 variants are displayed for participants in each serogroup, defined by binding antibody reactivity to spike (S) and nucleocapsid (N). Boxplots denote median IC50, and interquartile range (IQR) and the dots represent individual responses. The dotted line denotes the 1:100 IC50 threshold above which neutralising activity was considered to be present for all variants (Ancestral, B.1.1.7, B.1.617.2, BA.1, B.A.2[14]. **b** Negative log10 adjusted *p* values comparing NAb

responses to each variant across serogroups (larger circles = smaller adjusted *p* values) using Kruskal–Wallis test. In cases in which Kruskal–Wallis testing indicated significant differences, post hoc testing using Dunn's test was performed. Correction for multiple comparisons was performed using the Bonferroni–Holm method. Adjusted *p* values below 0.05 (red), 0.05−0.09 (orange) and >0.09 (blue) are highlighted. The lowest serum dilution screened for neutralisation was 1:50, with samples not neutralising at this concentration allocated random values between 1 and 49 for the purpose of statistical analysis and data visualisation, *n* = 342.

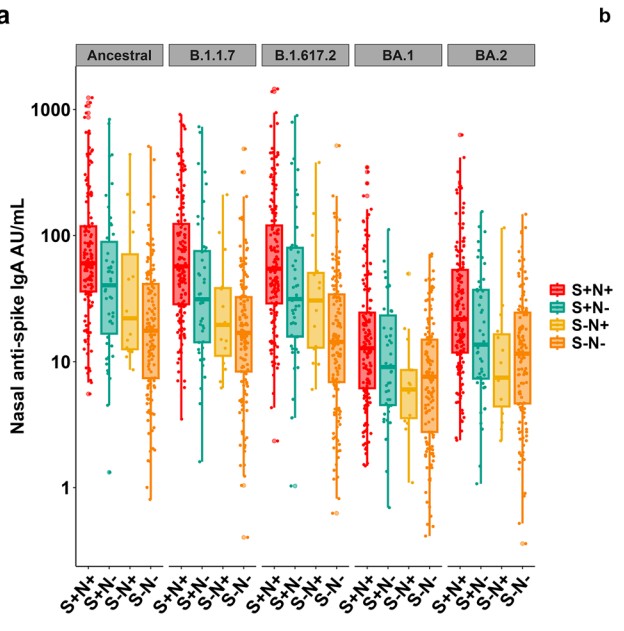

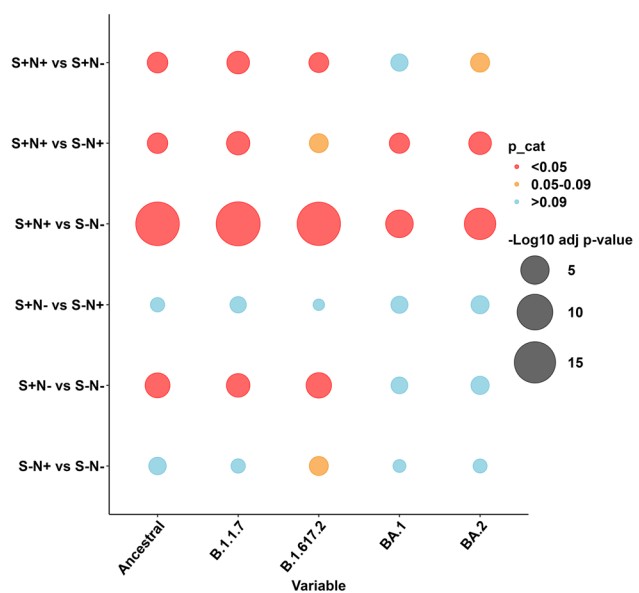

**Fig. 3 | Mucosal IgA responses to SARS-CoV-2 variants. a** Binding antibody units (AU/mL) to Ancestral, Alpha (B.1.1.7), Delta (B.1.617.2), Omicron BA.1 and BA.2 variants is displayed for participants in each serogroup, defined by binding antibody reactivity in serum to spike (S) and nucleocapsid (N). Boxplots denote median AU/mL and interquartile range (IQR), and the dots represent individual responses.
**b** Negative log10 adjusted *p* values comparing IgA responses to each variant across

serogroups (larger circle = smaller adjusted *p* values) using Kruskal–Wallis test. In cases in which Kruskal–Wallis testing indicated significant differences, post hoc testing using Dunn's test was performed. Correction for multiple comparisons was performed using the Bonferroni-Holm method. Adjusted *p* values below 0.05 (red), 0.05−0.09 (orange) and >0.09 (blue) are highlighted, *n* = 347.

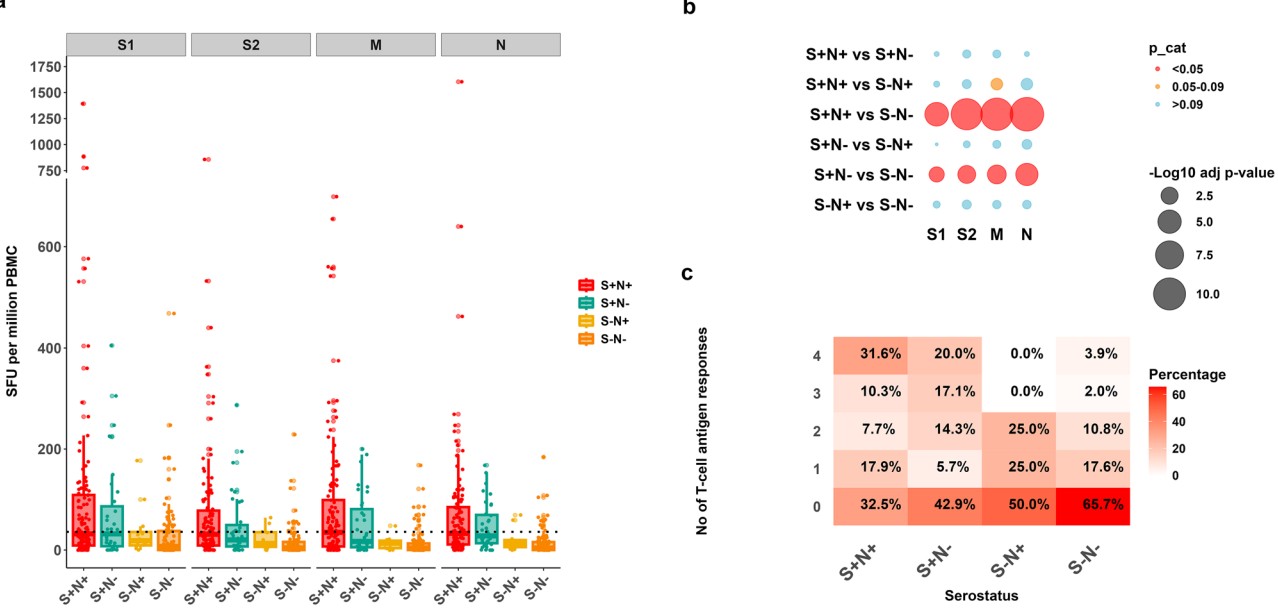

**Fig. 4 | T-cell responses to SARS-CoV-2 antigens.** Interferon-gamma ELISpot responses expressed as spot-forming units per million peripheral blood mono-nuclear cells (PBMC; SFU/mL). **a** T-cell responses to spike S1 and S2 subunits, membrane (M) and nucleocapsid (N) peptide pools are displayed for participants in each serogroup, defined by binding antibody reactivity in serum to spike (S) and nucleocapsid (N). Boxplots denote median SFU/ml PBMC responses and IQR, and the dots represent individual responses. **b** Negative log10 adjusted p values comparing T-cell responses to each antigen across serogroups (larger circles = smaller adjusted p values) using Kruskal–Wallis. In cases in which Kruskal–Wallis testing indicated significant differences, post hoc testing using Dunn's test was performed. Correction for multiple comparisons was performed using the Bonferroni–Holm method. Adjusted p values below 0.05 (red), 0.06–0.09 (orange) and ≥0.1 (blue) are highlighted. **c** Proportion of participants within each serogroup with T-cell responses to 0, 1, 2, 3 or 4 peptide pools tested, n = 266.

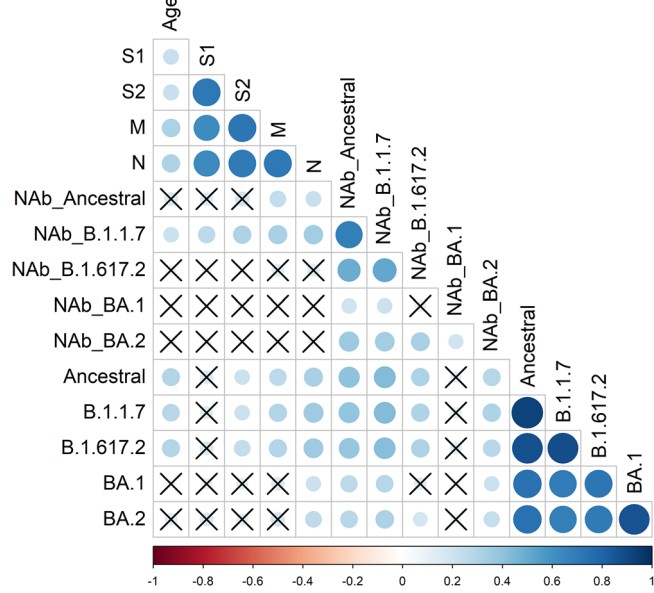

**Fig. 5 | Correlation between SARS-CoV-2 T-cell, serum neutralising antibody, and spike-specific mucosal IgA responses.** Correlation matrix depicting Spearman correlation coefficients (R) between each immune response, with adjusted p values using the Holm test. The size of the circle and the colour intensity relate to the absolute value of the Spearman rank correlation coefficient. The legend shows the correlation coefficients and the corresponding colours. Positive correlations are coloured blue. Correlations with p values > 0.05 are marked with an X. Compared are serum neutralising antibody (NAb) responses and mucosal IgA responses to Ancestral, Alpha (B.1.1.7), Delta (B.1.617.2), Omicron BA.1 and BA.2 variants, along with interferon-gamma T-cell responses to peptide pools representing spike S1 and S2 subunits, membrane (M) and nucleocapsid (N) proteins.

comparing T-cell, serum NAb, mucosal and IgA responses (Fig. 5). A weak positive correlation was observed between age and SARS-CoV-2 T-cell responses, and mucosal specific responses to the ancestral, Alpha and Delta strains.

Strong positive correlations were observed between all T-cell pools and between IgA responses to different SARS-CoV-2 variants (Fig. 5). Furthermore, positive correlations were present between NAb responses to ancestral, Alpha and Delta variants. Correlations between immune compartments were absent or weak, showing compartmentalised immune memory to SARS-CoV-2.

## Discussion

Within this comprehensive household study conducted in a peri-urban setting in The Gambia, we have established the high seroprevalence present early in the SARS-CoV-2 pandemic, and neutralising antibody, mucosal and T-cell responses to SARS-CoV-2 in adults and children. In pre-pandemic samples, there was a higher seroprevalence of nucleocapsid-specific SARS-CoV-2 antibody responses in adults compared to young children. Whilst strong correlation was noted within each immune parameter, the correlations between the different immune parameters were weak, suggesting differences in the cellular, systemic and mucosal immune responses elicited against SARS-CoV-2.

Following the first two waves of SARS-CoV-2 and prior to the emergence of the Delta variant, high SARS-CoV-2 seroprevalence was observed among unvaccinated, mostly asymptomatic adults individuals living in a peri-urban setting in The Gambia[8]. Similar findings were seen in an earlier study from Farafenni a partly rural and partly urban settlement in The Gambia, though that study focused solely on pregnant women and measured SARS-CoV-2 total receptor binding domain (RBD) immunoglobulin (Ig) M/IgG[15].

Our study revealed noticeable age-specific differences in seroprevalence rates[16]. When adjusted for spike protein only, we found that just

25% of children under 5 years old were seropositive, compared to over 70% of participants in the 18–49-year age group. Higher SARS-CoV-2 seroprevalence has been observed among younger and middle-aged adults in various African settings, albeit with significant variations[17–20], and in high-income settings such as the UK[21,22]. The differences in SARS-CoV-2 seroprevalence across Africa, may be influenced by country-specific and regional factors, along with variations in behaviour and pandemic measures, which likely impacted transmission dynamics and may explain the observed regional disparities[23].

Neutralising serum antibody responses were higher in 5–17-year age group compared to adults aged 18–49 years for the ancestral variant. Increased neutralising antibody titres in young children compared to adults have been previously described[24]. The high neutralising antibodies in these young children may arise from repeated viral infections, especially to seasonal coronaviruses which are cross-reactive against SARS-CoV-2[25]. Given the wide circulation of coronaviruses, especially amongst children, we accessed pre-pandemic samples to measure potential cross-reactive antibody to spike and nucleocapsid protein. The nucleocapsid-specific responses to human coronaviruses are conserved and highly cross-reactive whilst the spike-specific responses are more diverse. Our findings of high nucleocapsid-specific responses to SARS-CoV-2, in pre-pandemic samples have also been reported in other African countries[26] and has been attributed to exposure to other seasonal coronavirus but also with the high burden of malaria[27] and dengue fever[28] in this region.

Mucosal IgA responses have been shown to protect against viral infections. An induction of broad and persistent mucosal antibody response after primary SARS-CoV-2 infection against the spike and RBD of the virus has already been described[29]. In these SARS-CoV-2-infected individuals, spike and RBD mucosal antibodies increased within 7–9 days after the start of symptoms and correlated with a lower viral load and faster resolution of systemic symptoms. We showed that participants infected with SARS-CoV-2 (S+N+) had higher mucosal responses compared to those who were seronegative for all variants tested. Whilst we did not see any significant differences in mucosal antibody responses with age in this cross-sectional study, we cannot definitively conclude that there is lower mucosal IgA induction in children. Children younger than 5 years in our cohort appear to be relatively protected against infection, as previously reported[30]. Compared with adults, we also have very low numbers of children at baseline, who are spike antibody positive, which is a limitation. The timing of infection in relation to our sampling time point is not known, making it difficult to compare antibody responses, considering antibody half-life. However, previous studies have reported an earlier and more robust induction of IgA responses in asymptomatic children compared to symptomatic children and adults[31]. The early induction of mucosal IgA in asymptomatic children may possibly control viral replication, leading to mild infections. A significant correlation in mucosal responses for Ancestral, Alpha, Delta and Omicron variants was noted in our cohort, suggesting that mucosal responses could be more cross-reactive compared to serum antibody responses. Although we have not assessed the durability of the mucosal antibody responses, detection of mucosal antibodies up to 9 months post infection has been reported[29]. Results from a human challenge model of SARS-CoV-2, showed that in adults both mucosal and systemic antibodies are detected within 10 days after infection, however, mucosal antibodies plateau at day 14 post infection whilst systemic antibodies continue to rise[32].

We were able to assess T-cell responses to SARS-CoV-2 antigens in a subset of participants and found that among participants who were seropositive for both spike and nucleocapsid, 67.5% had a T-cell response to at least one antigen whilst 49.6% responded to two or more antigens. Interestingly, we also observed that 50% of (S−N+) and 34.3% of (S−N−) individuals had a T-cell response to at least one SARS-CoV-2 antigen. SARS-CoV-2-specific T-cell responses in spike-seronegative individuals has also been reported in Kenya with 70% of these individuals responding to multiple SARS-CoV-2-specific peptide pools as opposed to responses in single peptide pools when T-cell responses were assessed in pre-pandemic

samples[33]. The difference in magnitude and breadth of SARS-CoV-2-specific T cells between pre-pandemic and exposed seronegative healthcare workers in the UK has also been reported[34]. Whilst the observed T-cell responses in seronegative participants in our study cohort may warrant further investigations, the observation that these responses were not limited to a single peptide, similar to the Kenyan study[33] suggests that they are genuine responses.

Cross-reactivity to other human coronaviruses, malaria[27] and viral infections such as dengue fever[28] and CMV[35] has been implicated in the detection of SARS-CoV-2-specific T cells in both spike and nucleocapsid seronegative individuals. Higher frequencies of SARS-CoV-2-specific T cells were found in CMV seropositive donors compared to CMV seronegative individuals using PBMCs collected pre-pandemic. Further analysis of these cross-reactive T cells showed that HLA-B: 35:01, which is the most common HLA-B epitope in The Gambian population[36] presents both the SARS-CoV-2 spike peptide FVSNGTHWF (FVS) and the unrelated CMV pp65 peptide IPSINVHHY (IPS)[35]. CMV seropositivity is very high in The Gambia, with over 85% of individuals being positive by the first year of life[37]. It will therefore be difficult to ascertain the exact contribution of CMV infection to the SARS-CoV-2 cross-reactive T cells in seronegative donors in our setting.

Abortive infections may be another explanation for the presence of SARS-CoV-2 T cells in seronegative participants. In seronegative participants, Swadling et al showed that SARS-CoV-2 T cells that respond to replication-transcription complex (RTC) epitopes could also recognise human coronaviruses, and these were associated with abortive infections in SARS-CoV-2 exposed seronegative donors[34]. This suggest that SARS-CoV-2 viruses that infect these seronegative individuals were most likely rapidly cleared by the highly conserved RTC specific T cells from prior infection with human coronaviruses. Swadling et al showed the presence of T-cell responses to both the structural and RTC region of SARS-CoV-2 in seronegative healthcare workers, but a higher frequency of these T cells was directed against the RTC in seronegative healthcare workers compared to unexposed individuals from a pre-pandemic cohort and those infected with SARS-CoV-2. NSP-12 and NSP13 of the RTC region were among the most conserved across SARS-CoV-2 clades, and NSP-12 was shown to be the region that elicited the highest average magnitude and frequency of responders without triggering humoral immune responses[34].

The presence of SARS-CoV-2-specific CD4+ and CD8+ T cells is linked to reduced severity of COVID-19 during active infection[38]. Emergence of SARS-CoV-2 variants is of great concern as it can lead to immune escape, but majority, of SARS-CoV-2-specific CD4+ T cells (93%) and of CD8+ T-cell (97%) in naturally infected or Pfizer/BioNTech and Moderna COVID-19 mRNA vaccinated individuals are conserved across variants including Alpha, Beta and Gamma strains[39]. Slower waning of SARS-CoV-2-specific T cells compared to antibodies had been previously described[40]. Whilst sustained CD4+ T and CD8+ T-cell responses up to 8 months post infection were reported in a study by Dan et al, a more recent study reported a decrease in the magnitude of these SARS-CoV-2-specific T-cell 10 months post infection with the ancestral SARS-CoV-2 strain[41]. The findings that induction of CD4+ and CD8+ T cells in a cohort of seronegative SARS-CoV-2 exposed individuals was associated with time since exposure[42], suggest that in our cohort, viral infection with SARS-CoV-2 may have occurred at a lower threshold without seroconversion in these seronegative individuals.

Overall, we saw weak correlations between age and most antigens tested in mucosal IgA and T-cell assays, whereas this observation was not so consistent between age and neutralising antibody responses. At the time of sampling, it is likely that participants would only have experienced infection with a single variant (ancestral or alpha). Of note, a correlation between age and neutralising antibody responses to alpha was observed, which was dominant in the SARS-CoV-2 wave immediately before sampling[8]. Both mucosal IgA and T-cell responses are likely to demonstrate greater cross-reactivity than neutralising antibody responses. Sex differences in responses

to SARS-CoV-2 have been previously reported[43,44]. We saw no significant differences in T-cell and mucosal antibody responses between males and females. Neutralising antibody responses to ancestral and alpha strains were, however, significantly higher in females compared to males ($p = 0.03$) (Supplementary Fig. 5).

The assessment of systemic, mucosal, and T-cell responses in the same individuals is a strength of our study, as our data have shown that systemic antibody responses to infection differ from those at the site of pathogen entry, the mucosa. Analysis of samples from both blood and mucosal sites should be incorporated in future studies to increase our understanding of compartmentalised immune responses to infection. Our study also strengthens the evidence that seroprevalence data based on antibody responses alone may underestimate the true burden of SARS-CoV-2 in populations. We reported that whilst 43.1% of the individuals in our study had detectable antibody responses to both spike and nucleocapsid, 67.5% had a T response to at least one antigen within these seropositive individuals, suggesting that T-cell responses may be more suitable in identifying seropositive individuals. For T-cell assays to be more widely used as epidemiological tools, further assay standardisation and minimisation of cross-reactivity would be desirable.

Our study has some clear limitations. The ideal serological definition of seropositivity would be positivity for both nucleocapsid and spike. However, due to the substantial pre-pandemic cross-reactivity observed with Nucleocapsid, we used Spike alone to determine seropositivity. Given Spike's high specificity, the likelihood of including false-positive cases is, however, minimal. We could not assess SARS-CoV-2-specific T-cell responses in the pre-pandemic samples due to sample unavailability, but the presence of SARS-CoV-2 spike, membrane, ORF3a, or ORF7 responses was detected in 31% of individuals tested using pre-pandemic samples[33]. We show pre-existing antibody and T-cell responses to coronaviruses, though the specific strain cannot be determined within our study. Future studies should assess antibody and T-cell responses against the four major human coronaviruses species (a-hCCC-NL63, a-hCCC-229E and b-hCCCHKU1, b-hCCC-OC43) in pandemic samples to clarify their role in pre-existing immunity. Another limitation of our study is the lack of flow cytometry data to assess polyfunctional T cells, which are known to increase with repeated SARS-CoV-2 exposure[45,46]. Finally, we were limited by the fact that we did not have a rural population with pandemic data to compare with our data. However, this has been addressed by the study done by Janha et al 2023, where data from both rural and urban settings in The Gambia have been reported[15].

In summary, our data shows high seroprevalence of SARS-CoV-2 in The Gambia after two waves of the virus, especially in adults. Infection with SARS-CoV-2 induced systemic, mucosal and T-cell responses, with significant correlation observed with earlier variants as opposed to the later variants. Variant-specific responses were highly correlated at the mucosal level but not the serum antibody responses.

## Data availability

The source data for all figures contained within the paper and the supplementary data can be found in the file Supplementary Data 1. Detailed clinical data cannot be publicly shared due to ethical restrictions and the potential for identifying included individuals. To request detailed clinical data access, please contact the corresponding authors, Ya-Jankey Jagne (ya-jankey.jagne@lshtm.ac.uk) and Beate Kampmann (beate.kampmann@charite.de), who will respond to the request. Upon approval, data can be made available through a data-sharing agreement.

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

## Acknowledgements

The study was funded by a United Kingdom Research and Innovation Grant (no. MC_PC_19084). We are extremely grateful to all the TransVIR participants who took part in the study. We would like to acknowledge the TransVIR field team members Malang Mbenga, Fansu Dibba, Yusupha Fadera, Yusupha Faal, Dawda Jawara, Omar Jallow, Momodou Lamin Sanyang, Sarjo Wassa Koita, Hulaimatu Bangura, Mary Gibba, Maryama Jawara and Fatou Sanyang for their dedication and hard work during the field study. Our gratitude also goes to Nfamara Camara from the Research Support Office for all his support with the project and Ebrima Marenah and Ibrahim Lette for data entry.

## Author contributions

S.J., R.D.W., M.D., A.K.S., A.K., T.I.d.S. and B.K. conceived and design the study. Y.J.J., D.J., A.D., M.D., N.B., M.G., R.D.W., S.J., E.L.S., F.T., A.F.T., T.I.d.S., and B.K. collected and processed laboratory data. Y.J.J., D.J., A.D., M.D., M.K., N.B., R.D.W., S.J., D.Z., M.N., J.S., A.S., H.H., B.L., A.K.S., N.T., A.K., D.H., T.I.d.S. and B.K. analysed and interpreted data. Y.J.J., D.J., A.D., N.B., H.H., M.D., M.K., T.I.d.S. and B.K. assessed and verified underlying data. Y.J.J., A.D., D.J. T.I.d.S. and B.K. drafted the article. All authors read and approved the final draft.

## Competing interests

The authors declare no competing interests.

## Additional information

**Ya Jankey Jagne** [ID][1,7] ✉**, Dawda Jobe** [ID][1,7]**, Alansana Darboe**[1,7]**, Madikoi Danso**[1]**, Natalie Barratt**[2,3]**, Marie Gomez**[1]**,**
**Rhys Wenlock** [ID][1]**, Sheikh Jarju**[1]**, Ellen Lena Sylva**[1]**, Aji Fatou Touray**[1]**, Fatoumata Toure**[1]**, Michelle Kumado**[1]**, Anja Saso**[1]**,**
**Domen Zafred**[2]**, Martin Nicklin**[2]**, Jon Sayers**[2]**, Hailey Hornsby** [ID][2,3]**, Benjamin Lindsey** [ID][2,3]**, Abdul Karim Sesay**[1]**, Nigel Temperton**[4]**,**
**Adam Kucharski**[5]**, David Hodgson**[5]**, Thushan de Silva** [ID][1,2,3,8] **& Beate Kampmann**[1,6,8] ✉

[1]Vaccines and Immunity Theme, Medical Research Council Unit, The Gambia at the London School of Hygiene and Tropical Medicine, Fajara, The Gambia. [2]Division of Clinical Medicine, School of Medicine and Population Health, The University of Sheffield, Sheffield, UK. [3]The Florey Institute of Infection, The University of Sheffield, Sheffield, UK. [4]Viral Pseudotype Unit, Medway School of Pharmacy, University of Kent, Canterbury, UK. [5]Centre for Mathematical Modelling of Infectious Diseases, London School of Hygiene and Tropical Medicine, Keppel Street, London, UK. [6]Charité Centre for Global Health; Charité Universitätsmedizin, Berlin, Germany. [7]These authors contributed equally: Ya Jankey Jagne, Dawda Jobe, Alansana Darboe. [8]These authors jointly supervised this work: Thushan de Silva, Beate Kampmann. ✉e-mail: ya-jankey.jagne@lshtm.ac.uk; beate.kampmann@charite.de

