## [Transparent Peer Review file · Communications Medicine]

Compartmentalised mucosal and blood immunity to SARS-CoV-2 associated with high seroprevalence before Delta wave in Africa

Corresponding Author: Dr YA JANKEY JAGNE

Version 0:

Reviewer comments:

Reviewer #1

(Remarks to the Author)

In this study conducted by De Silva T. and Kampmann B., the seroprevalence and immune response to SARS-CoV-2 in The Gambia were analyzed after the first two pandemic waves, before the emergence of the Delta and Omicron variants and before the introduction of vaccines. The study included 349 participants from 52 households, and the key findings are as follows:

1. Seroprevalence: Overall seroprevalence was 56.7%, with the highest rate in adults (78%) and the lowest in children under 5 years (26.2%).
2. Neutralizing Antibodies: Seropositive individuals exhibited stronger responses, particularly against the Alpha variant. Children (5-17 years) showed stronger responses to the ancestral strain compared to adults.
3. Mucosal IgA Responses: Higher responses were observed in seropositive individuals, indicating broad cross-reactivity with various SARS-CoV-2 variants.
4. T-cell Immunity: Detected in many seropositive individuals and some seronegative ones, suggesting possible cross-reactivity or abortive infections.

The study emphasizes the significance of both systemic and mucosal immunity in understanding SARS-CoV-2 responses. It also suggests that T-cell responses may be more indicative of past infections than antibodies alone. Pre-existing immunity from other coronaviruses may have influenced these immune responses. While the study is well-executed and the manuscript is well-crafted, there are a few minor concerns that should be addressed before publication.

Specific comments:

1. Seroprevalence: The relatively low specificity of the ELISA for anti-N IgG (84.1%) significantly contributes to the observed cross-reactivity. The authors appropriately adjusted the seroprevalence estimates based on reactivity to the S protein alone. It would be beneficial for the authors to include this in the limitations of this study.
2. Mucosal IgA Responses: Authors may further explain in the discussion section their observation of no significant age-related differences in their cohort, which contrasts with more studies that have reported more robust mucosal IgA responses in asymptomatic children.
3. T-cell immunity: The authors suggest in the discussion that T-cell responses may be more suitable in identifying seropositive individuals. In such a scenario, they may also refer to the fact that T-cell assays are generally more complex and less standardized than antibody tests. Furthermore, such assays must be designed to minimize cross-reactivity and improve specificity for SARS-CoV-2, as extensively described in the manuscript.

Reviewer #3

(Remarks to the Author)

Jagne et al describe in this study the cellular and humoral immune response in blood and nasal mucosa to SARS-Cov2 in the Gambia during the pandemic prior to delta appearance. They find high seroprevalence in the study participants with antibody specificity not only against perental and alphas strains but also delta. Although T-cell responses were significantly higher in seropositive individuals, many sero negative participants showed SARS Cov2 specific T cell responses. To date, there are relatively few studies that have mapped immune responses to the pandemic in Africans. However, such studies are important as they could help to understand the observation the number of SARS-CoV-2 cases and deaths are lower in Africa compared to many high-income countries. As such this work is timely and highly relevant. The study is well executed and the interpretation and discussion of the presented data is balanced and fair. I only have a couple of minor suggestions to further improve the manuscript

- 1) It would be of greatly added value to compare differences in responses between males and females, given the growing appreciation in immunological differences in responses to infection between sexes.
- 2) The authors show that there is a correlation between age and T cell responses and mucosal ab responses, but not with circulating ag specific ab responses. Can the authors provide their view on how they interpret this?
- 3) For Fig 2 the font size is too small. This needs to be increased.
- 4) It would be important to show data of corrected seroprevalence in main Fig1, not as supplemental data
- 5) Half of fig 5 can be removed as it is a duplication of the other half

Version 1:

Reviewer comments:

Reviewer #1

(Remarks to the Author)

In this revised version of the manuscript, the authors have successfully addressed all of my comments and concerns regarding the seroprevalence and immune response to SARS-CoV-2 in The Gambia. This study provides valuable insights into the immune response to SARS-CoV-2 in a unique population and timeframe. The revised manuscript is now comprehensive and scientifically robust, rendering it suitable for publication.

Reviewer #3

(Remarks to the Author)

The authors have sufficiently addressed my concerns and I have no further comments

Dear Reviewers,

We thank you for the thorough review of our paper and herewith like to resubmit it for your kind attention.

We are grateful to our reviewers for their constructive and helpful comments, which we believe have now been fully addressed and have served to further improve our manuscript. Below, we provide the replies to each comment and how/where we have addressed them in the revised manuscript.

We trust that the replies and corrections will now allow this version to be accepted.

With best wishes,

Dr Ya-Jankey Jagne and Professor Beate Kampmann

Reviewer #1 (Remarks to the Author):

1. Seroprevalence: The relatively low specificity of the ELISA for anti-N IgG (84.1%) significantly contributes to the observed cross-reactivity. The authors appropriately adjusted the seroprevalence estimates based on reactivity to the S protein alone. It would be beneficial for the authors to include this in the limitations of this study.

Thank you – we have now additionally acknowledged this as a limitation 530-534.

2. Mucosal IgA Responses: Authors may further explain in the discussion section their observation of no significant age-related differences in their cohort, which contrasts with more studies that have reported more robust mucosal IgA responses in asymptomatic children.

We are limited by the fact that we do not know exactly when participants in the study were infected. We also have very few children at baseline who were spike positive, limiting our comparison with adults. We have explained this in Lines 429-435.

3. T-cell immunity: The authors suggest in the discussion that T-cell responses may be more suitable in identifying seropositive individuals. In such a scenario, they may also refer to the fact that T-cell assays are generally more complex and less standardized than antibody tests. Furthermore, such assays must be designed to minimize cross-reactivity and improve specificity for SARS-CoV-2, as extensively described in the manuscript.

We have acknowledged the limitations of T cell assays in lines 526-528.

Reviewer 2: (Remarks to the Author):

1. It would be of greatly added value to compare differences in responses between males and females, given the growing appreciation in immunological differences in responses to infection between sexes.

Thank you for pointing out this important point - we have added this analysis in line 511-515 and added a supplementary figure (S5)

2. The authors show that there is a correlation between age and T cell responses and mucosal ab responses, but not with circulating antigen specific ab responses. Can the authors provide their view on how they interpret this?

Overall the correlations are weak for age and IgA and T cell data. Both of these are also more likely to be cross-reactive across variants than the neutralizing antibody data. We have attempted an explanation in line 505-510.

“Overall, we saw weak correlations between age and most antigens tested in mucosal IgA and T-cell assays, whereas this observation was not so consistent between age and neutralizing antibody responses. At the time of sampling, it is likely that participants would only have experienced infection with a single variant (ancestral or alpha). Of note, a correlation between age and neutralizing antibody responses to alpha was observed, which was dominant in the SARS-CoV-2 wave immediately before sampling. Both mucosal IgA and T-cell responses are likely to demonstrate greater cross-reactivity than neutralizing antibody.”

3) For Fig 2 the font size is too small. This needs to be increased.

Thank you for highlighting this – we have redrawn the figure and increased the font size lines 295. We have additionally done this for figure 3, for consistency line 318-319.

4) It would be important to show data of corrected seroprevalence in main Fig 1, not as supplemental data.

We agree with this suggestion and have now incorporated this as part of figure 1. Line 251-252 and 261-262 figure 1 c.

5) Half of fig 5 can be removed as it is a duplication of the other half
Thank you – we have replaced this with a new figure. Line 370-371.